# Efficient Facial Landmark Localization Based on Binarized Neural Networks

**Hanlin Chen [1,†], Xudong Zhang [2,†], Teli Ma [3], Haosong Yue [1,*], Xin Wang [4,*] and Baochang Zhang [1]**

[1]    School of Automation Science and Electrical Engineering, Beihang University, Beijing 100191, China; hlchen@buaa.edu.cn (H.C.); bczhang@buaa.edu.cn (B.Z.)

[2]    School of Electronics and Information Engineering, Beihang University, Beijing 100191, China; 15231009@buaa.edu.cn

[3]    ShenYuan Honors College, Beihang University, Beijing 100191, China; mtl9868@buaa.edu.cn

[4]    Shenzhen Academy of Aerospace Technology, Shenzhen 518000, China

*    Correspondence: yuehaosong@buaa.edu.cn (H.Y.); xin.wang@chinasaat.com (X.W.)

†    These authors contributed equally to this work.

**Abstract:** Facial landmark localization is a significant yet challenging computer vision task, whose accuracy has been remarkably improved due to the successful application of deep Convolutional Neural Networks (CNNs). However, CNNs require huge storage and computation overhead, thus impeding their deployment on computationally limited platforms. In this paper, to the best of our knowledge, it is the first time that an efficient facial landmark localization is implemented via binarized CNNs. We introduce a new network architecture to calculate the binarized models, referred to as Amplitude Convolutional Networks (ACNs), based on the proposed asynchronous back propagation algorithm. We can efficiently recover the full-precision filters only using a single factor in an end-to-end manner, and the efficiency of CNNs for facial landmark localization is further improved by the extremely compressed 1-bit ACNs. Our ACNs reduce the storage space of convolutional filters by a factor of 32 compared with the full-precision models on dataset LFW+Webface, CelebA, BioID and 300W, while achieving a comparable performance to the full-precision facial landmark localization algorithms.

**Keywords:** facial landmark localization; amplitude convolutional networks (ACNs); binarized neural networks (BNNs)

---

## 1. Introduction

Facial localization is a fundamental computer vision task [1] which lays the foundation for many higher level applications, including face verification [2], facial emotion recognition [3], human–computer interaction [4], and facial motion capture [5]. Having demonstrated their superior power in numerous vision tasks [6], deep Convolutional Neural Networks (CNNs) have also exhibited major performance boost in solving facial landmark localization, which significantly outperform the traditional hand-crafted predecessors. However, considering the complex and deep nature of CNNs, their success in facial landmark localization comes with prohibitive computation and storage overhead. Although they perform well on expensive GPU-based machines, they are often unsuitable for resource-constrained devices like cell phones and embedded electronics [7]. Accordingly, how to implement these successful CNN-based solutions for resource-limited platforms remains unsolved. Considering the rapid developments of portable smart devices and the growing demand of facial recognition, there is a pressing need to overcome the resource limitations and realize the state-of-the-art

deep facial landmark localization solutions on portable devices. To this end, substantial research efforts have been devoted to design light-weight CNNs by compressing redundant parameters of convolutional kernels [8,9]. One typical network compression strategy is to approximate floating-point kernel weights with binary values [10–12]. Following this line of research, local binary convolution (LBC) layers are introduced in [13] to binarize the non-linearly activated responses of standard convolutional layers. In [12], a BinaryConnect scheme using real-valued weights as a key reference is exploited to improve the binarization process. In [10,11], XNOR-Net is presented where both the weights and inputs of the convolution kernels are approximated with binary values, to improve the efficiency of the convolutional operations. In [14,15], DoReFa-Net exploits 1-bit convolution kernels with low bit-width gradients to accelerate both the training and inference phases. ABC-Net [15] adopts multiple binary weights and activations to approximate full-precision weights. More recently, a simple fixed-scaling method incorporated in 1-bit convolutional layers is proposed to binarize CNNs, resulting in results that are comparable with full-precision networks [16]. Modulated convolutional networks (MCNs) are introduced in [17] to binarize the kernels, which however are not real 1-bit CNNs, achieving better performance compared with full-precision baselines.

In this paper, we introduce a simple yet efficient facial landmark localization algorithm via binarized CNNs. A new Amplitude Convolutional Network (ACN) is proposed to achieve computation efficiency for 1-bit CNNs. In our ACN, the full-precision kernels and amplitude matrices are obtained during back propagation to minimize the information loss caused by kernel binarization in the forward pass. The process is called asynchronous back propagation, which can efficiently recover the full-precision filters using only one single factor in an end-to-end manner. Optimized with the asynchronous back propagation algorithm, our ACN not only saves the storage by a factor of 32, but also achieves comparable landmark localization performance, compared with its corresponding full-precision model. The contributions of this paper are summarized as follows:

(1)　We propose a new Amplitude Convolutional Network (ACN) for facial landmark localization, to achieve computation efficiency for resource constrained mobile devices, using binarized CNNs.
(2)　We design a new asynchronous back propagation algorithm to optimize ACNs efficiently, leading to an extremely compressed 1-bit CNNs model in an end-to-end manner.
(3)　ACN achieves comparable facial landmark localization performance compared with its corresponding full-precision CNN model on several benchmark datasets including CelebA, BioID, LFW and Webface.

## 2. Related Work

Facial landmark localization. Facial landmark localization has a long history, from the early Active Appearance Models [18,19] to Constrained Local Models [20,21], and has recently shifted to methods based on Cascaded Shape Regression [22–28] and deep learning [29–33].

In particular, representative deep facial landmark localization methods usually adopt cascaded CNNs. For instance, Sun et al. [34] constructed a cascaded multi-level regression network with a deep CNN to detect facial points. Differently, Zhou et al. [29] proposed a facial landmark localization algorithm following the coarse-to-fine strategy and achieved high-precision positioning of 68 face key points. Thereafter, Zhang et al. [6] proposed a multi-task Cascaded Convolutional Network (MTCNN) to simultaneously deal with face detection and facial landmark localization. Notably, these facial landmark localization methods with cascaded CNNs all employ patch-wise operations, wherein the original input images are processed patch-by-patch. This strategy is time consuming and suffers from the boundary effects. As an improvement, Kowalski et al. [1] proposed a new cascaded deep neural network, named Deep Alignment Network (DAN), which avoids the drawbacks of patch-wise operations and processes an input images as a whole. More recently, Liu et al. [35] detected facial landmarks in not only the spatial domain, but also exploited the temporal domain using recurrent neural networks (RNNs) in video-based face alignment.

In summary, the deep CNN based methods discussed above mainly focus on solving a cascaded regression problem with iterative optimization. Such solutions are resource-intense and computation-heavy, therefore prohibit the deployment on resource-constrained devices such as mobile phones.

Network binarization. Several methods attempt to binarize the weights and the activations in neural networks.The performance of highly quantized networks (e.g., binarized) were believed to be very poor due to the destructive property of binary quantization. Expectation BackPropagation (EBP) in [36] showed high performance can be achieved by a network with binary weights and binary activations In [12], a BinaryConnect scheme using real-valued weights as a key reference is exploited to improve the binarization process. The work of [11] goes one step further and binarizes both parameters and activations. In this case multiplications can be replaced with elementary binary operations In [10], XNOR-Net is presented where both the weights and inputs of the convolution kernels are approximated with binary values via reconstructing the unbinarized filters simply by the average of absolute weight values, leading to a bad performance compared with full-precision version. Ref. [10] is the first work to report good results on a large dataset (ImageNet). Instead of focusing on image classification, Ref. [37] propose a novel binary architecture network in the facial landmark localization task and enhance the performance but using the same method for binarization as in XNOR-Net [10]. Our method differs from all aforementioned works, it is the first time that an efficient facial landmark localization is implemented by improving the quantization method.

## 3. Amplitude Convolutional Networks

### 3.1. Problem Formulation

The inference process of any binary neural network (BNN) model is based on the binarized kernels, which means that the kernels must be binarized in the forward step (corresponding to the inference) during training, so that the training loss is calculated based on the binarized filters. Accordingly, we have to update the kernel $x$ using the gradient derived from the binarized kernel, i.e.,

$$x^{k+1} \leftarrow x^k - \eta \delta_x^k, \tag{1}$$

where $\delta_x^k$ is calculated based on the binarized kernel $\hat{x}$, which is elaborated in Section 3.4. Contrary to the forward process, during back propagation, the resulting kernels are not necessary to be binarized and can be full-precision. Under this situation, the full-precision kernels are binarized to gradually bridge the binarization gap during training. Therefore, the learning of most BNN models involve both discrete and continuous spaces, which poses great challenges in practice. To address these challenges and improve the optimization of binarizing CNNs, we decouple the kernel $x$ and represent $\hat{x}$ by its amplitude and direction as:

$$\hat{x} = a \cdot d, \tag{2}$$

where $a$ and $d$ respectively denote the amplitude and the direction of $x$. $d$ is easily calculated by $sign(x)$ as $-1$ for negative $x$ and $1$ for positive $x$. We can then focus on the computation of the amplitude $a$, which involves complex kernel reconstruction and feature learning. To compute $a$, we propose an asynchronous back propagation method.

### 3.2. Loss Function of ACNs

In order to achieve binarized weights, we design a new loss function in ACNs. Note that the kernels of ACNs are binarized, while for 1-bit ACNs, both the kernels and the activations are binarized, which will be briefly described at the end of Section 3.4. In Table 1, we elaborate $D$, $A$ and $\hat{A}$: $D_i^l$ are the directions of the full-precision kernels $X_i^l$ of the $l^{th}$ convolutional layer, $l \in \{1, \cdots, N\}$; $A^l$ shared by all $D_i^l$ represents the amplitude of the $l^{th}$ convolutional layer; $\hat{A}^l$ and $A^l$ are of the same size and all the elements of $\hat{A}^l$ are equal to the average of the elements of $A^l$. In the forward pass, $\hat{A}^l$ is used

instead of the full-precision $A^l$. In this case, $\hat{A}^l$ can be considered as a scalar. The full-precision $A^l$ is only used for back propagation during training. This process is the same as the way of calculating $\hat{x}$ from $x$ in an asynchronous manner, which is also illustrated in Figure 1. Accordingly, Equation (3) is represented for ACNs as:

$$\hat{X} = \hat{A} \odot D, \tag{3}$$

where $\odot$ denotes the element-wise multiplication between matrices. We then define an amplitude loss function to reconstruct the full-precision kernels as:

$$L_{\hat{A}} = \frac{\theta}{2} \sum_{i,l} \| X_i^l - \hat{X}_i^l \|^2 = \frac{\theta}{2} \sum_{i,l} \| X_i^l - \hat{A}^l \odot D_i^l \|^2, \tag{4}$$

where $D_i^l = sign(X_i^l)$ represents the binarized kernel. The element-wise multiplication combines the binarized kernels and the amplitude matrices to approximate the full-precision kernels. Our final loss function is defined with the MSE loss:

$$L_S = \frac{1}{2S \cdot M} \sum_{s,m} \| \hat{Y}_{s,m} - Y_{s,m} \|_2^2, \tag{5}$$

where $\hat{Y}_{s,m}$ is the ground truth of the $m^{th}$ facial landmark for the $s^{th}$ example; $Y_{s,m}$ is the corresponding detected position by ACNs. Finally, the overall loss function $L$ is applied to supervise the training of ACNs in the asynchronous back propagation in the following form:

$$L = L_S + L_{\hat{A}}. \tag{6}$$

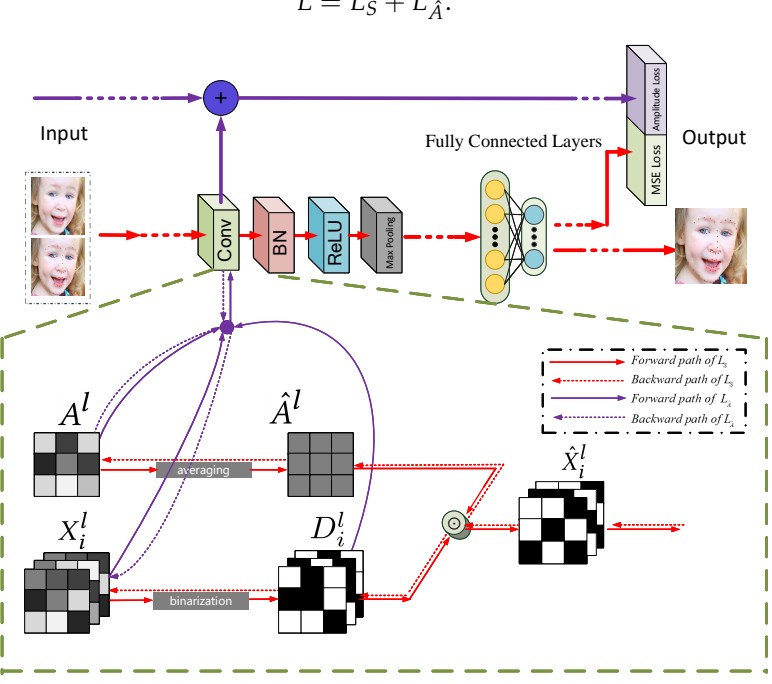

**Figure 1.** Facial landmark localization based on Amplitude Convolutional Networks (ACNs). ACNs are designed based on the convolutional kernels' directions $D_i^l$ and amplitude matrices $A^l$. The amplitude matrices are particularly designed to efficiently approximate the unbinarized convolutional filters in the end-to-end framework. Since an amplitude matrix is shared at each layer, ACNs are high compressed. The full-precision convolutional kernels $X_i^l$ are quantized to $D_i^l$ via the sign function. Both the new loss $L_{\hat{A}}$ and the traditional loss $L_S$ are used to train ACNs. We illustrate our network structure based on conventional Convolutional Neural Networks (CNNs), and more specific details are shown in the dotted box. $\odot$ indicates the element-wise multiplication between matrices. Note that the amplitude matrices $A^l$ and the full-precision kernels $X_i^l$ are not used in the inference. ACNs have binarized kernels. To compress CNNs, 1-bit ACNs are proposed based on ACNs by further binarizing the full-precision feature maps of ACNs into binary feature maps.

**Table 1.** A brief description of the main notations used in the paper.

| | | |
|---|---|---|
| $X$: full-precision kernel | $\hat{X}$: binarized kernel | $A$: amplitude matrix |
| $F$: feature map | $D$: $X'$s direction | $\hat{A}$: generated from A |
| $i$: kernel index | $g$: input feature map index | $h$: output feature map index |
| $m$: facial landmark index | $l$: layer index | $M$: number of facial landmarks |
| $S$: number of examples | | |

*3.3. Forward Propagation of ACNs*

In ACNs, $\hat{X}^l$ in the $l^{th}$ layer are used to calculated the output feature maps $F^{l+1}$ as:

$$F^{l+1} = ACconv(F^l, \hat{X}^l), \tag{7}$$

where *ACconv* denotes the newly designed amplitude convolution operation in Equation (8). A simple illustration of the forward pass operation in ACNs is depicted in Figure 1. In ACconv, the channels of the output feature maps are generated as follows:

$$F_h^{l+1} = \sum_{i,g} F_g^l \otimes \hat{X}_i^l, \tag{8}$$

where $\otimes$ denotes the convolution operation; $F_h^{l+1}$ is the $h^{th}$ feature map in the $(l+1)^{th}$ convolutional layer; $F_g^l$ denotes the $g^{th}$ feature map in the $l^{th}$ convolutional layer.

*3.4. Back-Propagation Updating*

In ACNs, what need to be learned and updated are the full-precision kernels $X_i$ and amplitude matrices $A$. The kernels and the matrices are jointly learned. In each convolutional layer, ACNs update the full-precision kernels and then the amplitude matrices. In what follows, the layer index $l$ is omitted for simplicity.

3.4.1. Updating the Full-Precision Kernels

We denote $\delta_{X_i}$ as the gradient of the full-precision kernel $X_i$, and have:

$$\delta_{X_i} = \frac{\partial L}{\partial X_i} = \frac{\partial L_S}{\partial X_i} + \frac{\partial L_{\hat{A}}}{\partial X_i}, \tag{9}$$

$$X_i \leftarrow X_i - \eta_1 \delta_{X_i}, \tag{10}$$

where $\eta_1$ is a learning rate. We then have:

$$\frac{\partial L_S}{\partial X_i} = \frac{\partial L_S}{\partial \hat{X}_i} \cdot \frac{\partial \hat{X}_i}{\partial X_i} = \frac{\partial L_S}{\partial \hat{X}_i} \cdot \hat{A} \cdot \mathbb{1}, \tag{11}$$

$$\frac{\partial L_{\hat{A}}}{\partial X_i} = \theta \cdot (X_i - \hat{A} \odot D_i), \tag{12}$$

where $X_i$ is the full-precision convolutional kernel corresponding to $D_i$, and $\mathbb{1}$ is the indicator function [10] widely used to estimate the gradient of non-differentiable function.

3.4.2. Updating the Amplitude Matrices

After updating $X$, we update the amplitude matrix $A$. Let $\delta_A$ be the gradient of $A$. According to Equation (6), we have:

$$\delta_A = \frac{\partial L}{\partial A} = \frac{\partial L_S}{\partial A} + \frac{\partial L_{\hat{A}}}{\partial A}, \tag{13}$$

$$A \leftarrow |A - \eta_2 \delta_A|, \tag{14}$$

where $\eta_2$ is another learning rate. Note that the amplitudes are always set to be non-negative. We then have:

$$\frac{\partial L_S}{\partial A} = \sum_i \frac{\partial L_S}{\partial \hat{X}_i} \cdot \frac{\partial \hat{X}_i}{\partial \hat{A}} \cdot \frac{\partial \hat{A}}{\partial A} = \sum_i \frac{\partial L_S}{\partial \hat{X}_i} \cdot D_i, \tag{15}$$

$$\frac{\partial L_{\hat{A}}}{\partial A} = \frac{\partial L_{\hat{A}}}{\partial \hat{A}} \cdot \frac{\partial \hat{A}}{\partial A} = -\theta \cdot (X_i - \hat{A} \odot D_i) \cdot D_i, \tag{16}$$

where $\frac{\partial \hat{A}}{\partial A}$ is set to 1 for easy implementation of the algorithm. Note that $\hat{A}$ and $A$ are respectively used in the forward pass and the backward propagation in an asynchronous manner. The above derivations show that ACNs are learnable with the new BP algorithm. The 1-bit ACNs are obtained via binarizing the kernels and activations simultaneously as in [10,11]. We summarize the training procedure in Algorithm 1.

---

**Algorithm 1** Optimization of ACNs with the asynchronous back propagation

---

**Require:**
    The training dataset; the full-precision kernels $X$; the amplitude matrices $A$; the learning rates $\eta_1$ and $\eta_2$.
**Ensure:**
    The ACNs with the learned $X$, $A$.
 1: Initialize $X$ and $A$ randomly;
 2: **repeat**
 3:    // Forward propagation
 4:    **for** $l = 1$ to $L$ **do**
 5:        Generate $\hat{A}^l$ from $A^l$; //Every element of $\hat{A}^l$ is the average of all the elements of $A^l$.
 6:        $\hat{X}_i^l = \hat{A}^l \odot \text{sign}(X_i^l), \forall i$;
 7:        **if** 1 bit **then**
 8:            $\hat{F}^l = \text{sign}(F^l)$; //1 bit ACNs
 9:        **end if**
10:        Perform 2D convolution with $\hat{X}_i^l, \forall i$;
11:    **end for**
12:    // Backward propagation
13:    Compute $\frac{\partial L_S}{\partial X_i^l}, \forall l, i$;
14:    **for** $l = L$ to $1$ **do**
15:        Calculate $\delta_{X_i^l}, \delta_A^l$; // using Equations (9)~(16)
16:        Update parameters $X_i^l$ and $A^l$ using Adam;
17:    **end for**
18: **until** the algorithm converges.

---

## 4. Implementation and Experiments

Our ACNs and 1-bit ACNs are generic, which can be applied to any CNNs architectures. In our experiments, we used four convolutional layers and two fully-connected (FC) layers as the backbone of our ACNs. Similar CNN structures are widely used for facial landmark localization in prior work [29,34,38–40]. Figure 2 shows the details of the network architectures of the backbone CNNs, ACNs and 1-bit ACNs. The input was the face region obtained from a face detector. The compared networks were evaluated using the same training and testing sets. To further evaluate the performance of our method, we first experimented on three small-scale datasets to verify the generalization ability of the model. Specifically, we trained the detectors only based on the LFW+Webface training dataset [29]. Then we further compared them on two other public test sets, CelebA and BioID, without changing the training set. Then a significant experiment on the large-scale benchmark 300-W was employed to compare with the state-of-the-art works.

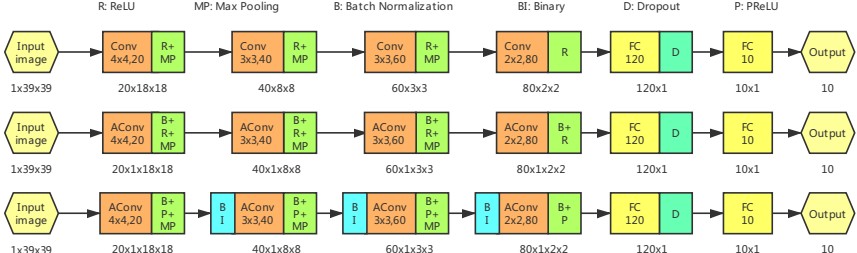

**Figure 2.** Network architectures of CNNs (top), Amplitude Convolutional Networks (ACNs) (middle, binarized kernels) and 1-bit ACNs (bottom).

### 4.1. Datasets and Evaluation Metric

**Datasets:** The LFW+Webface dataset was processed by [34] and it was composed of a training set of 10,000 and a testing set of 3466 face images, among which 7876 images were downloaded from the web and the remaining 5590 images are from LFW. Each face was labeled with the positions of five keypoints.

The CelebA dataset [41] was a large-scale face attribute dataset with more than 200 K celebrity images, each with 40 attribute annotations. The images in the dataset covered large pose variations and background clutter. CelebA had large diversities and quantities, and rich annotations. Each image was also labeled with the positions of five keypoints. Therefore, it was convenient for us to test the model on the dataset. Finally, we randomly chose 2599 images as the testing set for the performance comparison.

The BioID dataset [42] consisted of 1521 gray level images of 23 persons with a resolution of $384 \times 286$ pixels. Each image showed the frontal view of a face. Each face was labeled with the positions of five key points by [34]. This test set was characterized by large variations in lighting, background, and face size. We used all images as our test images and compared the performance of different algorithms.

300-W [43] consisted of several datasets including AFW, HELEN, LFPW, XM2VTS. 300 W had a 68-point annotation for each face image, We followed [44] to split the dataset into four sets, training, common testing, challenging testing, and full testing, respectively. We regarded all the training samples from LFPW, HELEN and the full set of AFW as the training set, in which there is 3148 training images. The full test set (689 images) was divided into a common subset (554 images), which contained the test sets from LFPW and HELEN, and a challenging subset (135 images) which was from IBUG.

**Evaluation metric.** We used the normalized mean error (NME) as the evaluation metric, which is defined as follows:

$$NME = \frac{1}{N \cdot d} \sum_{i=1}^{N} \sqrt{(\hat{x}_i - x_i)^2 + (\hat{y}_i - y_i)^2} \tag{17}$$

where $(x, y)$ and $(\hat{x}, \hat{y})$ denotes the ground truth and predicted coordinates respectively, $N$ denotes the number of landmarks, and for LFW+Webface, CelebA and BioID datasets, $d$ is the face size, for 300-W dataset, $d$ is the inter-ocular distance. For brevity, % is omitted in all tables.

### 4.2. Implementation Details

In our experiments, four NVIDIA GeForce TITAN X GPUs are used. On LFW+ Webface, CelebA and BioID datasets, all the original images were firstly cropped by face bounding box and resized to $39 \times 39$ gray-scale. We replaced the spatial convolution layers with ACconv modules. In all the experiments, we adopted Max-pooling except the last convolution layer, ReLU after the convolution layers, and a dropout layer after the first FC layer to avoid over-fitting. We used the Adam optimization algorithm [45]. $\beta_1$ and $\beta_2$ were the exponential decay rates for the first and the second moment estimates and they were set to 0.9 and 0.999 respectively. The two learning rates $\eta_1$ and $\eta_2$ in Equations (10) and (14) were both set to $1 \times 10^{-4}$. The models were trained for 1600 epochs. In addition, in Equation (4), $L_{\hat{A}}$ was balanced by the hyper-parameter $\theta$, which was empirically set to $5 \times 10^{-4}$.

On 300-W dataset, we utilized CPM [46] as our backbone, which extracted features by the first four convolutional layers of VGG-16 and predicted heatmaps with three CPM stages. Based on the ImageNet pre-trained model, we replaced the CPM convolution with ACconv modules. We follow the settings of [46] that randomly cropped and resize faces to (256, 256). The model was trained with SGD for 40 epochs using batch size of 8. The learning rate was set to $5 \times 10^{-5}$ at first and reduced by 0.5 at 20th and 30th epochs.

All the models were implemented using the PyTorch platform. We plotted the training error curves of CNN, ACN and 1-bit ACN in Figure 3. The curves showed that the convergence of ACN, marginally affected by the binarization process, was similar to the corresponding full-precision CNN. This validated the accuracy and reliability of the model.

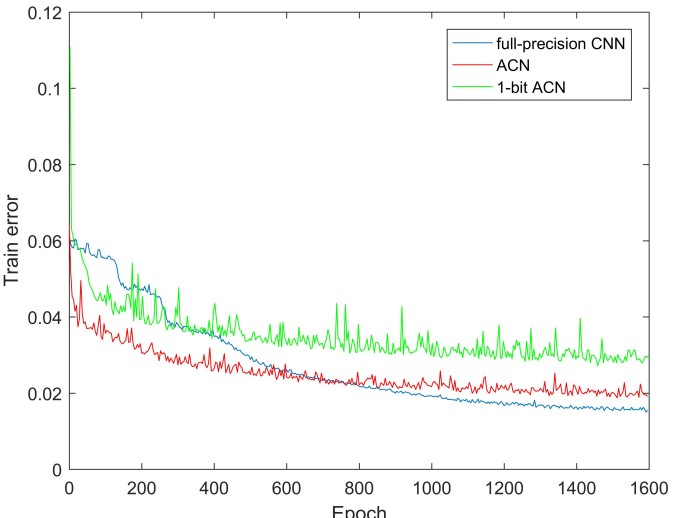

**Figure 3.** Training error curves.

*4.3. Results on LFW+Webface, CelebA and BioID*

In this section, we first compare the full-precision CNN, ACN, and 1-bit ACN with four kernel structures, 10-20-40-80, 20-40-60-80, 40-60-80-100, and 20-20-40-40-60-60-80-80. The structures shown in Figure 2 correspond to 20-40-60-80.The experimental results are given in Table 2 with the normalized mean error(NME) and amounts of model parameters. Next, we describe the results on all the datasets.

**LFW+Webface:** From Table 2, it is observed that for the kernel structure 20-40-60-80, ACN achieved 2.1925% error on the test set, which is comparable to the full-precision CNN (2.06%). Although the detection error 3.0633% of the 1-bit ACN was a little higher than the full-precision CNN, this difference can be ignored with the significant increase of computational efficiency. When the kernel structure became 40-60-80-100, ACN achieved 1.9429% error, which outperformed the full-precision CNN (2.0190%).

**CelebA**: We tested three networks on the CelebA dataset, which were trained on the LFW+Webface training set. In Table 2, for the kernel structure 20-40-60-80, ACN obtained 2.1538% error, which is very close to the full-precision CNN (2.1256%). At the same time, the gap between the 1-bit ACN and the full-precision CNN was only 0.3811%. When the kernel structure becomes 40-60-80-100, ACN achieved 1.9091% error, which again outperformed the full-precision CNN (2.0608%). Moreover, the detection error of the 1-bit ACN reduced when the network became deeper.

**BioID**: With the three networks trained on the LFW+Webface training set, we compared them on BioID. The errors of the full-precision CNN, ACN, and 1-bit ACN reached 1.669%, 1.7769%, and 2.4555% respectively, for the kernel structure 20-40-60-80. The slight difference between the full-precision CNN and ACN further validated the effectiveness of our model. Meanwhile, the 1-bit ACN maintained a close performance to the full-precision CNN. When the kernel structure became

40-60-80-100, ACN yielded 1.6064% error, which again performed better than the full-precision CNN (1.675%).

**Table 2.** Results comparison of normalized mean error (NME) on different datasets. Note that for each kernel structure, the numbers of the parameters of the three networks are similar. However, ACN and 1-bit ACN use binarized kernels.

| Network Kernels | #Param. | Model | LFW+Webface | CelebA | BioID |
|---|---|---|---|---|---|
| 10-20-40-80 | 0.06M | full-precision CNN | 2.1532 | 2.2503 | 1.7286 |
| | | ACN | 2.2436 | 2.2344 | 1.8395 |
| | | 1-bit ACN | 3.1224 | 2.7782 | 2.8510 |
| 20-40-60-80 | 0.09M | full-precision CNN | 2.0600 | 2.1256 | 1.6690 |
| | | ACN | 2.1925 | 2.1538 | 1.7769 |
| | | 1-bit ACN | 3.0633 | 2.5067 | 2.4555 |
| 40-60-80-100 | 0.15M | full-precision CNN | 2.0190 | 2.0608 | 1.6750 |
| | | ACN | 1.9429 | 1.9091 | 1.6064 |
| | | 1-bit ACN | 2.8099 | 2.4284 | 2.4298 |
| 20-20-40-40-60-60-80-80 | 0.16M | full-precision CNN | 1.8994 | 2.0194 | 1.4668 |
| | | ACN | 1.9000 | 2.1663 | 1.4045 |
| | | 1-bit ACN | 3.1091 | 2.9447 | 2.6785 |

A few visual detection results are presented in Figure 4. Besides, examples of the kernels and feature maps of the three networks are illustrated in Figures 5 and 6. From Figure 6, we can see that similar to the full-precision CNN, the features from different layers of ACN and 1-bit ACN captured the rich and hierarchical information, which demonstrated the advantages of our models as well as the binarized kernels.

**Failure cases.** To gain more insight into our proposed method, we show some challenging cases that failed our model in Figure 7. In the first three cases, the regions highlighted by the yellow circle were occluded, which made it challenging for them to be detected. In the last three cases, large head pose variations destroyed the face shape and led to poor performance, which could be relieved by using post processing operations. We will address these problems in the future work.

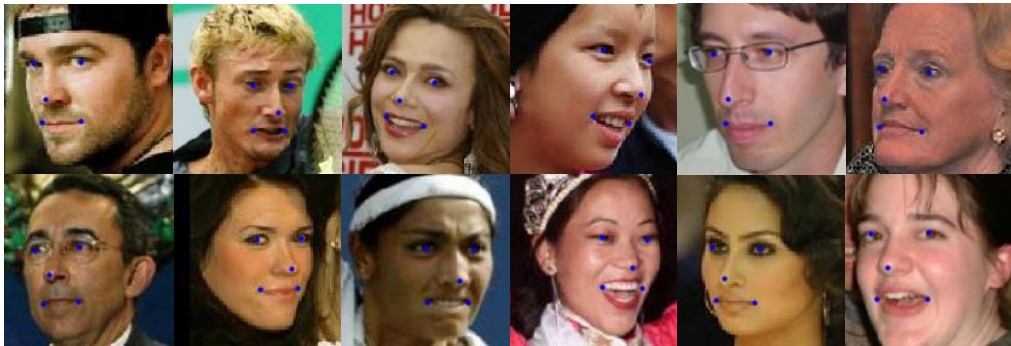

**Figure 4.** Some detection results of faces with large pose and expression variations. The face images are from LFW+Webface. First row: results by ACN. Second row: results by 1-bit ACN. Both of the two models use the 20-40-60-80 kernel structure.

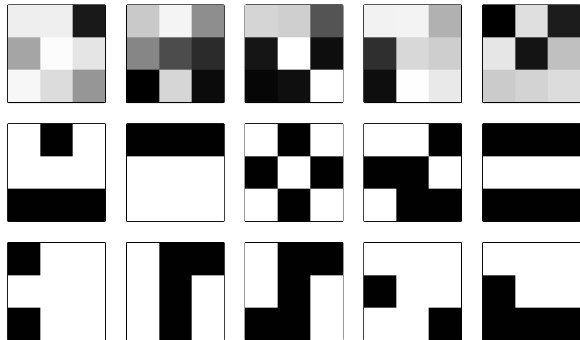

**Figure 5.** Illustration of kernels of the full-precision CNN (first row), ACN (second row) and 1-bit ACN (third row).

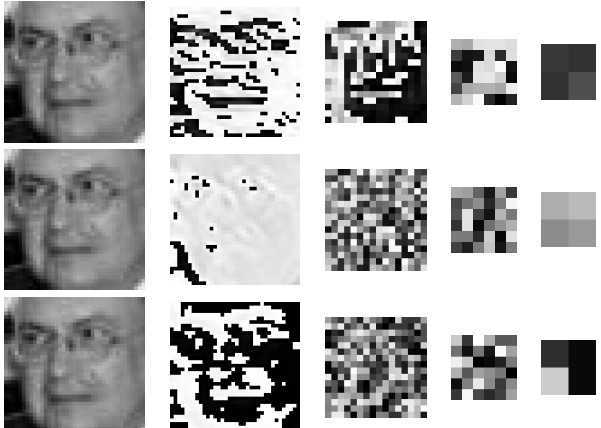

**Figure 6.** Illustration of the feature maps of the four layers of the full-precision CNN (first row), ACN (second row) and 1-bit ACN (third row). The network structures are shown in Figure 2. Note that similar to the full-precision CNN, the features from different layers of ACN and 1-bit ACN capture the rich and hierarchical information, which illustrates the advantages of our models.

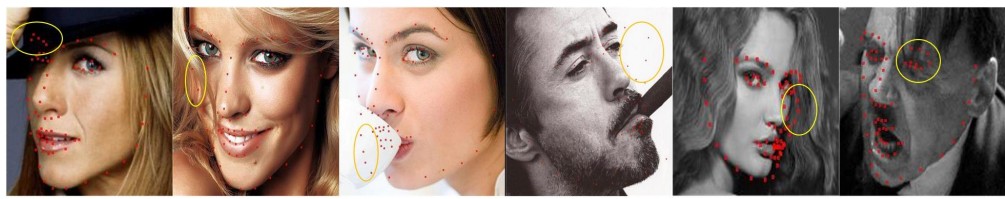

**Figure 7.** Visualization of some failure cases on the 300-W dataset. Yellow circles indicate the clear failures.

### 4.4. Results on 300-W

Table 3 shows the performance of different facial landmark detection algorithms on the 300-W and AFLW dataset. We compared our approach with recently proposed state-of-the-art algorithms. Our ACN was established following the settings in baseline method CPM [46] but replaced the convolution with ACconv modules. The 1-bit ACN was obtained via binarizing the kernels and activations simultaneously. The implementation details are in Section 4.2. As in shown, our ACN with binarized kernels achieved very competitive results 4.39 NME on the full set of 300-W and 2.28 NME on the AFLW compared with others. Note that our ACN gave better results on the challenging set of 300-W compared with the baseline method, which indicated the robustness of our method and that ACN could alleviate the overfitting problem by increasing the model compactness, since one amplitude matrix was shared at one layer and also the binarized kernels were used. Moreover, there was only a

small gap between the 1-bit ACN and the real-valued version of the baseline CPM. Note that our 1-bit ACN outperformed many current state-of-the-art methods, all of which use large real-valued CNNs. Figure 8 shows visual detection results of ACN and 1-bit ACN on 300-W dataset.

**Table 3.** Comparisons of NME with state-of-the-art methods on 300-W and AFLW datasets.

| Methods | 300-W | | | AFLW |
|---|---|---|---|---|
| | Common | Challenging | Full Set | |
| SDM [22] | 5.57 | 15.40 | 7.52 | 5.43 |
| LBF [24] | 4.95 | 11.98 | 6.32 | 4.25 |
| MDM [31] | 4.83 | 10.14 | 5.88 | - |
| TCDCN [38] | 4.80 | 8.60 | 5.54 | - |
| CFSS [44] | 4.73 | 9.98 | 5.76 | 3.92 |
| DSRN [47] | 4.12 | 9.68 | 5.21 | **1.86** |
| Two-Stage [48] | 4.36 | 7.56 | 4.99 | 2.17 |
| CPM(baseline) [46] | **3.39** | 8.14 | **4.36** | 2.33 |
| ACN | 3.67 | **7.36** | 4.39 | 2.28 |
| 1-bit ACN | 4.16 | 8.97 | 5.10 | 2.91 |

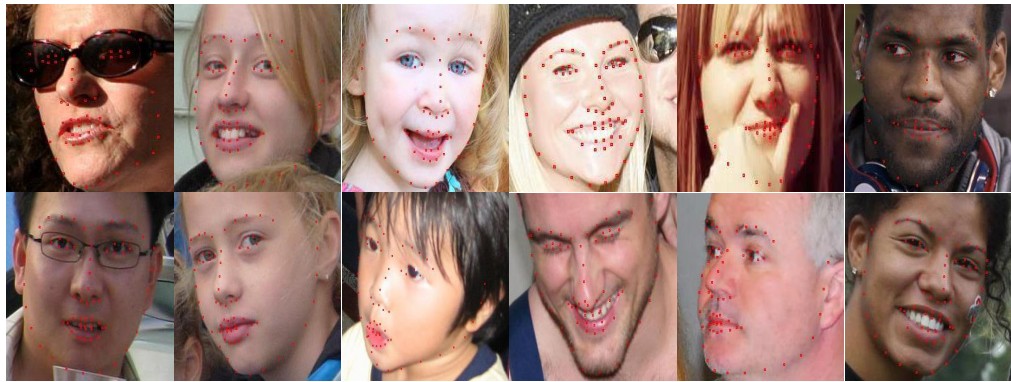

**Figure 8.** Qualitative results of our ACN (row 1) and 1-bit ACN (row 2) on 300-W dataset.

**Efficiency analysis:** The memory usage of a network is computed as the summation of 32 bits times the number of full-precision kernels and 1 bit times the number of the binary kernels in the network. As shown in Table 4, our ACN reduced the memory usage by 32 times compared with the full-precision CNN. The reason is that the amplitude matrices were only used when training for enriching the diversity of the kernels of ACN, whereas they were not used in inference. XNOR-Net [7] is another popular 1-bit CNN, and can also be used for the same landmark detection task. Note that the parameter amount of ACN was almost the same as XNOR-Net.However, the ways of approximating the unbinarized filters were clearly different, where the unbinarized filters in XNOR-Net were reconstructed using binary filters with a single scaling factor while ours accomplished it by a set of amplitude matrics together with the binary filters. For the efficiency analysis of the 1-bit ACN, if all the convolution operations were binary, then the convolutions could be estimated by XNOR and bit counting operations [11], which gained 58× speedup in CPUs same as XNOR-Net [7]. We compared XNOR-Net with our ACN and 1-bit ACN on the 300-W dataset.The results are given in Table 4, from which we observe that our models consistently outperformed XNOR-Net with a large margin in all the cases.

**Table 4.** Comparisons of performance with XNOR-Net on 300-W dataset. Note that the results of memory saving and speedup are compared with the full-precision baseline CPM.

| Method | Memory Saving | Speedup | Common | Challenging | Full Set |
|---------|-----------|-----------|----------|----------|----------|
| XNOR-Net | 32× | 58× | 5.97 | 12.43 | 7.24 |
| ACN | 32× | - | **3.67** | **7.36** | **4.39** |
| 1-bit ACN | 32× | 58× | 4.16 | 8.97 | 5.10 |

## 5. Conclusions

In this paper, we propose an efficient facial landmark localization framework, termed as binarized CNNs. Specifically, we improve the efficiency and compactness of CNNs for facial landmark localization via the extremely compressed 1-bit ACNs. On top of that, we also design a new architecture to recover the full-precision kernels efficiently using a single factor at each convolution layer in an end-to-end manner. Compared with the full-precision networks, our ACNs not only save the storage space by a factor of 32 but also achieve comparable localization performance. In addition, ACNs perform better than the popular binary CNNs, such as XNOR-Nets. Code is available at https://github.com/bczhangbczhang.

**Author Contributions:** Conceptualization, H.C., X.Z. and B.Z.; methodology, H.C. and X.Z.; software, H.C., X.Z. and T.M.; Validation, H.C. and X.Z.; formal analysis, H.Y., X.W. and B.Z.; investigation, H.C. and X.Z.; resources, H.Y., X.W. and B.Z.; data curation: H.C. and X.Z.; writing—original draft preparation, H.C., X.Z. and T.M.; writing—review and editing, H.Y., X.W. and B.Z.; visualization, H.C.; supervision, B.Z.; project administration, X.W. and H.Y.; funding acquisition, X.W. All authors have read and agreed to the published version of the manuscript.

**Funding:** This research was funded by National Natural Science Foundation of China grant number 61672079.

**Acknowledgments:** The work is in part Supported by Shenzhen Science and Technology Program (No.KQTD2016112515134654).

**Conflicts of Interest:** The authors declare no conflict of interest.

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
