# Peer review of "Efficient Facial Landmark Localization Based on Binarized Neural Networks"

_electronics, doi:10.3390/electronics9081236_

Round 1

Reviewer 1 Report

This manuscript proposes a method based on kernel binarisation and amplitude convolutional neural networks for higher efficiency in the task of facial landmark detection. The method is novel and shows potential for efficient deep learning in other tasks. However, the performance and efficiency advantages over the literature in face landmark detection are not clear, since the performance benchmark is weak and there is no comparison of efficiency with established state-of-the-art methods. 

Strengths: (1) novel method; (2) strong motivation; (3) widely applicable; (4) very good writing;

Weaknesses: (1) efficiency gains over the literature are not clear; (2) weak performance benchmarking with the literature.

  1. The abstract should include a quick overview of the databases used (only their names) and the NMS results obtained for each of them.

  1. In line 136, “Our ACNs and 1-bit ACNs are generic, which can be applied to any CNNs architectures.” Why do the authors focus on face landmark detection then? The paper would be appeal to a much larger crowd if it wasn’t so focused on this specific task. I am not asking for any change here, just stating that probably the method is being under-valued by the authors.

  1. Figure 4 is too small, detected landmarks are not really visible. The authors should enlarge the images and clearly denote the location of the predicted and ground-truth landmarks. Additionally, it would be interesting to have one such image for successful difficult cases, and another for unsuccessful cases, and a discussion on what went wrong.

  1. Tables 2 and 3 are valuable, but what about a comparison of the proposed method with literature approaches (like in Table 3) on the other databases (LFW+Webface, CelebA, and BioID)? Comparing on just one database looks like cherry-picking and is not enough to assert the superiority of the proposed method.

  1. As this paper is on efficient facial landmark localization, one would expect to see a run time per image and memory requirements comparison with the state-of-the-art methods. Table 4 only shows that the proposed method is lighter than one other simple alternative, not established literature methods like those mentioned in the related work.

  1. Related to the previous comment: The authors state they use 4x NVidia Titan X GPUs. Given that the method is supposed to be efficient, maybe using simpler settings (using GPU just for training, and only CPU for testing, for example) and presenting the run times would be very beneficial. 

  1. In the conclusion, the authors should delve a bit on future work proposals. What were the flaws of the presented method? What could be done to improve? What additional experiments do the authors want to explore next?

Reviewer 2 Report

Thank you for the opportunity to review this paper. In this paper the authors proposed a novel deep neural network architecture, Amplitude Convolutional Networks (ACNs). The paper is well written and easy to understand. The approach is sound and well described. There are only several minor things that the authors need to address, i.e.:

1. One of the main goals or contributions of this paper is to propose a new method that could achieve computation efficiency for resource constrained mobile devices (page 3). However, in the experiments, the hardware (4 NVIDIA GeForce TITAN X GPUs) doesn't seem to support what the authors want to achieve. Although the authors present an efficiency analysis (memory saving and speedup), but the efficiency analysis was not done in mobile devices specifically.

2. I think to say that the convergence of ACN was similar to the corresponding full-precision CNN (Fig. 3) was not convincing because the error margin is too different.

3. Please make the code available on public repository, e.g. Github for others to review it. The code should be prepared so others can reproduce similar results shown on the paper.

4. Table 4. Which kernel structure was used for the comparisons? Please provide this information.

5. Following query #4, please also provide the comparisons for all other kernel structures for each dataset. This is important to show if the efficiency should be sacrified in order to get a better performance.

Reviewer 3 Report

This paper introduces a simple variation to improve binarized neural networks for facial landmark localization. The experiments show that it could potentially be useful in low-resource devices. Two minor issues:

  1. The very first word in the introduction is a typo, please proofread the manuscript thoroughly.
  2. Please explain the term "asynchronous" more carefully, as it doesn't look obvious what is asynchronous here. Figure 1 depicts the parallel processing of A and D. That doesn't seem to be asynchronous. 

Author Response

Thank you for your review!

Point 1: The very first word in the introduction is a typo, please proofread the manuscript thoroughly.

Response 1: Thank you for your carefulness, we thoroughly checked spelling for another time.

Point 2: Please explain the term "asynchronous" more carefully, as it doesn't look obvious what is asynchronous here. Figure 1 depicts the parallel processing of A and D. That doesn't seem to be asynchronous. 

Response 2:Well, the term "asynchronous" depicts the process which the full-precision kernels and amplitude matrices are obtained during back propagation to minimize the information loss caused by kernel binarization in the forward pass. They are not parallel but asynchronous, since they may happen at different time.

Reviewer 4 Report

The authors confirmed that binary CNNs can be a good alternative to full-precision networks. Under the adopted assumptions, the proposed ACNs in both versions achieve efficiency comparable with state-of-the-art algorithms.
Some concern is only the resistance of the proposed approach, especially 1-bit ACN, to large variations in size and proportion of face bounding box occurring in real applications.
Additional tests for standardized face sizes larger than 39x39 could also be useful, which would show the generality of the proposed approach.
Some comment regarding large local changes in the learning error would be appreciated, especially for 1-bit ACN (Fig. 3).
Reporting the error with the precision of four decimal places (for a percentage) may seem professional, but it is hardly legible.
Minor Errors:
Zhang 2, † Teli Ma - no comma
Ficial
Generet
. in
5x10-5
CPM [45] - spaces are missing in several places
filters.For
are presented in Figs. 4
in Figs. 5 and 6

Round 2

Reviewer 1 Report

I thank the authors for answering most of my comments. I still have one minor and one major comment.

Regarding comment 1, I still believe it is important to state briefly the databases names and quantitative results in the abstract.

Regarding comment 3, I understand it may be difficult to obtain such images, but it is very important to disclose and discuss where the proposed algorithm succeeded and failed. Showing just successes is not acceptable. Given that you have the 300-W database divided into 'common' and 'challenging' sets, it shouldn't be difficult for the authors to scan it and find challenging images where the method succeeded (lower NME), and common/challenging images where the method failed (higher NME). This should be accompanied by a discussion of likely reasons and possible future improvements. 

Round 3

Reviewer 1 Report

I thank the authors for answering my comments and revising the manuscript accordingly. I believe it is now much clearer and complete.